# Efficacy and safety of different antidepressants and anticonvulsants in central poststroke pain: A network meta-analysis and systematic review

**Ke-Yu Chen**[1], **Ruo-Yang Li**[2]*

**1** Department of Traditional Chinese Medicine, Chengdu Second People's Hospital, Chengdu, Sichuan, China, **2** Hospital of Chengdu University of Traditional Chinese Medicine, Chengdu, Sichuan, China

☯ These authors contributed equally to this work.

* alis7718@outlook.com

**Data Availability Statement:** All relevant data are within the paper and its Supporting information files.

## Abstract

### Objective

To evaluate the efficacy and safety of different antidepressants and anticonvulsants in the treatment of central poststroke pain (CPSP) by network meta-analysis and provide an evidence-based foundation for clinical practice.

### Methods

PubMed, Cochrane Library, EMBASE, CNKI, APA PsycINFO, Wanfang, VIP and other databases were searched by computer to find clinical randomized controlled studies (RCTs) on drug treatment of CPSP. The retrieval time limit was from the establishment of each database to July 2022. The quality of the included RCTs was evaluated using the bias risk assessment tool recommended by Cochrane. Stata 14.0 was used for network meta-analysis.

### Results

A total of 13 RCTs, 1040 patients and 9 drugs were finally included. The results of the network meta-analysis showed that the effectiveness ranking as rated by the visual analog scale (VAS) was gabapentin > pregabalin > fluoxetine > lamotrigine > duloxetine > serqulin > amitriptyline > carbamazepine > vitamin B. Ranking according to the numerical rating scale (NRS) was pregabalin > gabapentin > carbamazepine. Ranking derived from the Hamilton depression scale (HAMD) was pregabalin > duloxetine > gabapentin > amitriptyline.

### Conclusion

All nine drugs can relieve the pain of CPSP patients to different degrees; among them pregabalin and gabapentin have the most significant effect, and gabapentin and pregabalin also have the most adverse reactions. In the future, more multicenter, large sample, double-blind

**Funding:** The author(s) received no specific funding for this work.

**Competing interests:** The authors have declared that no competing interests exist.

clinical randomized controlled trials need to be carried out to supplement and demonstrate the results of this study.

## Introduction

Stroke has the characteristics of a high incidence and high disability rate. CPSP, as one of the common sequelae after stroke, has always disrupted the daily life of patients. CPSP is focal, lesion-related pain that occurs continuously or intermittently in a part of the paralyzed body after hemorrhage or ischemic stroke and is accompanied by sensory abnormalities. Some studies have shown that CPSP has a certain latency period, mostly occurring within 3–6 months after stroke and, in some cases, even 18 months post-stroke [1, 2], and the occurrence of CPSP is closely related to sensory impairment. CPSP is related to the location of the lesion. Patients with brainstem and thalamic stroke are more likely to have central neuropathic pain, with a higher incidence of CPSP after lesions in the lateral medulla and lateral thalamus and a lower incidence of CPSP after lesions in other locations. The pathogenesis of CPSP is unclear, and scholars have proposed many hypotheses, among which the "central sensitization" and "disinhibition" hypotheses have received widespread attention. The "central sensitization" hypothesis suggests that damage to the sensory system is accompanied by changes in neurotransmitters, excitotoxicity, and inflammatory responses, leading to a loss of neuronal inhibition or excitatory function, resulting in "central sensitization" and chronic pain; the "disinhibition" hypothesis suggests that damage to sensory pathways leads to compensatory overactivation of the thalamus, resulting in spontaneous or abnormal pain [3–5].

Drug therapy is the most common intervention therapy for CPSP. Antidepressants and anticonvulsants are among the more common types of agents used in drug therapy. They all control CPSP episodes mainly by modulating transmitters such as adrenaline, 5-hydroxytryptamine, aminobutyric acid, and glutamatergic neurotransmitters [5], and the duration of CPSP pharmacotherapy in clinical practice depends on the patient's degree of pain relief, which is usually 4–8 weeks and, in some cases, even longer. At present, many clinical studies [6, 7] have shown significant efficacy, but few studies have compared the efficacy and safety among the various agents. Based on the frequency method in network meta-analysis, this study evaluated the clinical efficacy and safety of different drugs in treating CPSP, with the VAS score, NRS score, HAMD score and clinical adverse reactions as outcome indicators. The relevant efficacy was ranked based on the surface under the cumulative ranking (SUCRA) score.

## Materials and methods

### Registration

This meta-analysis has been registered on PROSPERO, registration number CRD42022352098.

### Literature search

Search databases: PubMed, Excel Medica database (EMBASE), Cochrane library, CNKI, APA PsycINFO, Wanfang, VIP. The retrieval time is from the establishment of the database to July 2022. The search terms included central poststroke pain, central poststroke pain, central neuropathic pain after stroke, central neuropathic pain after cerebrovascular disease, central

neuropathic pain after cerebral infarction, central neuropathic pain after intracerebral hemorrhage, central poststroke pain, central poststroke pain, and central poststroke pain. The combination of subject words and free words was used for retrieval. Taking PubMed as an example, its specific strategies are as follows:

1# (central poststroke pain[Title/Abstract])

2#(central post-stroke pain[Title/Abstract])

3#(central neuropathic pain after stroke[Title/Abstract])

4#(central neuropathic pain after cerebrovascular disease[Title/Abstract])

5#(central neuropathic pain after cerebral infarction[Title/Abstract])

6#(central neuropathic pain after intracerebral hemorrhage[Title/Abstract])

7#(central poststroke pain[MeSH Terms])

8#(central post-stroke pain[MeSH Terms])

9#(central post stroke pain[Title/Abstract])

10#(thalamic pain[Title/Abstract])

Search: 1# or 2# or 3# or 4# or 5# or 6# or 7# or 8# or 9# or 10#
See the appendix for all retrieval strategies.

## Study selection

The research type was RCTs. 2. The subjects were identified as stroke patients by brain CT, MRI and other imaging examinations and were diagnosed with CPSP with a pain score of 3 or above. 3. Treatment scheme: the control group was treated with placebo, a single antidepressant or a single anticonvulsant, and the treatment group was treated with a single antidepressant or a single anticonvulsant. 3. Outcome indicators: primary indicators: VAS; secondary indicators: NRS, HAMD.

## Study exclusion

Unrelated literature, repeated literature, review, case report, meta-analysis and other nonrandomized controlled trials; 2. Documents with repeated data, incomplete data or unavailable data; 3. Documents repeatedly published in Chinese and English; 4. Literature with unclear drugs in the control group.

## Data extraction

Two researchers independently performed literature screening and data extraction according to the inclusion and exclusion criteria and cross-checked the results. In case of disagreement, the third researcher participated in the discussion and made a decision. Documents extracted from the database were imported into Endnote, and manual and automatic duplicate checking was performed to eliminate duplicate documents. After the titles and abstracts of the literature were read, the inconsistent studies were initially removed, and the full text of the remaining literature was reviewed. The inconsistent studies were removed twice. For literature with incomplete data, the author was contacted to find the missing information. The two researchers separately extracted the requisite literature details, including patient information, intervention measures and outcome indicators. If there were differences, the third researcher judged them. All continuous data were included in the difference between before and after changes (i.e., the

difference between the indices after treatment and before treatment). If the original text was not calculated, it will be calculated by myself. The formula was as follows: corr is usually 0.5.

$$SD_{E.change} = \sqrt{SD^2_{E.baseline} + SD^2_{E.final} - (2 \times Corr \times SD_{E.baseline} \times SD_{E.final})}$$
$$Mean_{E.change} = Mean_{E.final} - Mean_{E.baseline}$$

## Methodologic quality

According to the quality evaluation scale recommended in Cochrane Handbook 5.1.0, the quality of the included literature was evaluated, including the generation from random number series, allocation concealment, whether to use the blind method, data integrity, selective reporting and other bias. The results were divided into low risk, high risk and unknown risk of bias. After the evaluation, two researchers conducted a cross check.

## Statistical analysis

This study is based on the framework of frequency science. For the continuity indicators VAS and NRS, the mean difference (MD) is adopted as the effect quantity due to the unified measurement method. For the continuity indicator HAMD, considering that there are three scales (HAMD-17, HAMD-21 and HAMD-24), to reduce the impact of different scales on the results, the standard mean difference (SMD) was adopted as the effect quantity, and the corresponding 95% confidence interval (CI) was calculated. Stata 14.0 software was used to draw the network evidence relationship map, forest map, grade probability map, funnel map and corresponding statistics. When testing the global consistency, if the difference is not statistically significant (i.e., $P > 0.05$), it indicates that there is no overall inconsistency [8]. The study calculated the inconsistency factors (if) and 95% CI of each closed loop in the network. This calculation method performs loop inconsistency detection by using the ifplot command in Stata. If the lower limit of the 95% confidence interval contained or was close to 0, it indicated that the local comparison, i.e., the direct comparison evidence, was consistent with the indirect comparison evidence. In addition, the node splitting method was used to perform the local consistency test. When $p > 0.05$, it suggests that the local inconsistency is not obvious. In this study, SUCRA was used to calculate the cumulative ranking probability of each treatment scheme. The larger the value of success was, the larger the area under the curve of the cumulative probability ranking graph, which indicates better effect of the intervention. This study followed the PRISMA extension for network meta-analysis.

## Results

### Included literature

Through the search, 2916 research studies were preliminarily obtained, including 329 CNKI, 680 Wanfang, 178 VIP, 597 PubMed, 683 EMBASE, 237 APA PsycINFO, and 212 Cochrane Library. They were entered into Endnote. After excluding duplicate studies and strictly following the nanodischarge criteria, 13 studies [9–21] finally met the inclusion criteria, including 1 three-arm trial [17], and the rest were two-arm trials, with a total of 1040 patients. A total of 9 drug interventions were involved, including pregabalin, gabapentin, lamotrigine, vitamin B, amitriptyline, serqulin, fluoxetine, carbamazepine and duloxetine. The document screening process and results are shown in Fig 1, and the basic information of the included documents is shown in Table 1.

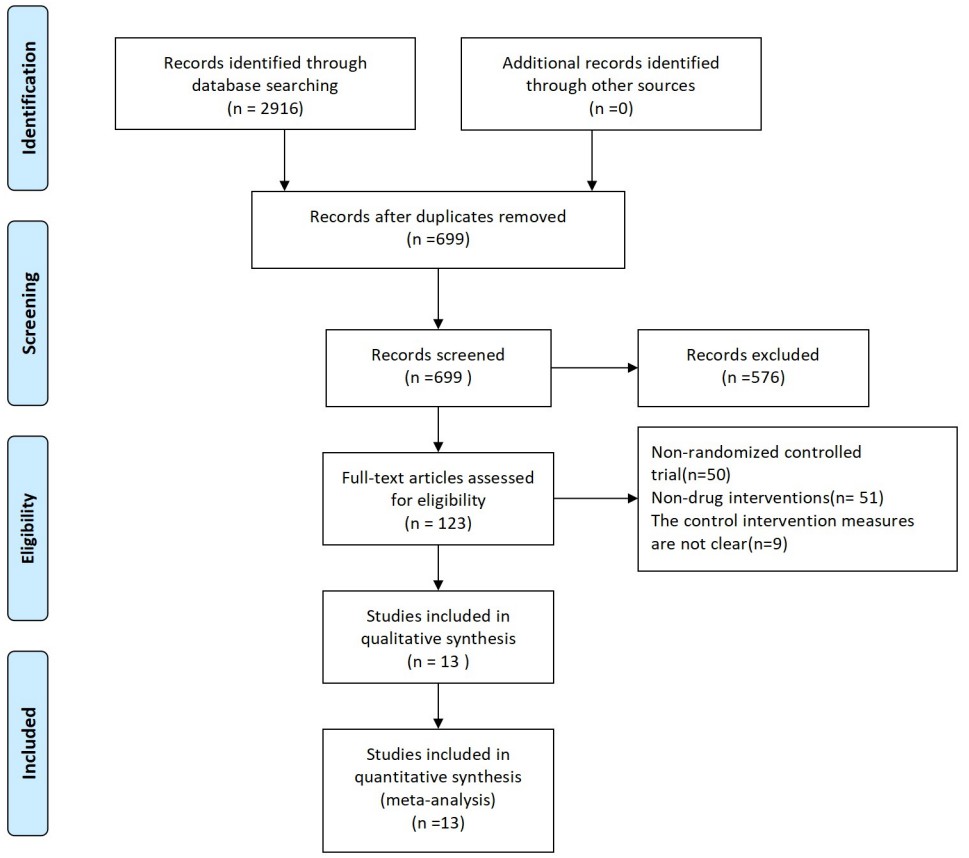

**Fig 1. Flow chart of literature screening.**

## Risk of bias

The included studies were evaluated using the Cochrane bias risk assessment tool. In terms of the random allocation method, 5 studies were low risk, and random allocation was conducted by the random number table method [9, 10, 14, 17, 21]. Two studies [11, 15] did not mention random allocation, which was high risk. The other studies only mentioned random allocation and did not report specific random allocation methods. In terms of the concealment of the random allocation scheme, one study [10] mentioned that the concealment of allocation was low risk, and the other studies did not mention it. In terms of the blind method of the random allocation scheme, one study [10] mentioned the use of the blind method, and the other studies did not mention it. The included literature on outcome measures was not mentioned. In terms of data integrity, the data included in the study were complete. In terms of selective reporting of research results, the included studies were all low risk. In terms of other sources of bias, the number of men and women in the patient base in one document [12] did not match the total number of included trials. The data was believed to have been entered incorrectly, so they were rated as high risk. The risk assessment of bias in the included studies is shown in Fig 2.

## Evidence network

A total of 10 studies reported the outcome indicator VAS, involving 9 drug treatment regimens. A total of 4 studies reported the outcome index NRS, involving 3 drug treatment regimens. A total of 4 studies reported the HAMD outcome index, involving 4 drug treatment

**Table 1. Basic characteristics of the included trials.**

| Author | Year | Number of patients | | | Age(year) | | | Male/female) | Treatment | | | Intervention period | Outcome indicator |
|---|---|---|---|---|---|---|---|---|---|---|---|---|---|
| | | I | C | | I | C | | | I | C | | | |
| Zhu HY [9] | 2013 | 32 | 31 | | 66.47 ±8.45 | 65.86±8.62 | | 40/23 | Pregabalin | Gabapentin | | 8W | NRS, HAMD, clinical adverse reaction |
| Kalita [10] | 2017 | 15 | 15 | | 53.7 ±11.6 | 53.9±10.4 | | 26/4 | Pregabalin | Lamotrigine | | 12W | VAS |
| Chen XY [11] | 2015 | 45 | 45 | | 67.1±8.3 | 67.6±8.5 | | 49/41 | Amitriptyline | Vitamin B | | 4W | VAS, clinical adverse reaction |
| Chen XD [12] | 2013 | 32 | 32 | | 63.8±6.5 | 63.1±5.6 | | - | Pregabalin | Carbamazepine | | 2W | VAS, clinical adverse reaction |
| Dai JC [13] | 2005 | 20 | 20 | | - | - | | - | Fluoxetine | Carbamazepine | | 4W | VAS |
| Gu C [14] | 2017 | 50 | 50 | | 64.61 ±5.22 | 64.62±5.23 | | 56/44 | Pregabalin | Gabapentin | | 4W | VAS, NRS, clinical adverse reaction |
| He GW [15] | 2015 | 75 | 75 | | 66.2±2.3 | 66.8±2.5 | | 83/67 | Lamotrigine | Pregabalin | | 8W | VAS, clinical adverse reaction |
| Huang C [16] | 2018 | 33 | 32 | | 69.35 ±4.25 | 68.44±4.31 | | 40/25 | Pregabalin | Amitriptyline | | 4W | VAS, HAMD, clinical adverse reaction |
| Liu C [17] | 2015 | 32 | 32 | 32 | 45–78 | 43–80 | 45–83 | 51/45 | Pregabalin | Gabapentin | Carbamazepine | 8W | NRS, clinical adverse reaction |
| Mai XL [18] | 2011 | 23 | 22 | | 59.3±6.0 | 58.8±6.2 | | 25/20 | Duloxetine | Amitriptyline | | 4W | VAS, HAMD |
| Yuan YJ [19] | 2016 | 36 | 36 | | 66.5±7.3 | 65.2±8.7 | | 37/35 | Gabapentin | Pregabalin | | 8W | NRS, HAMD, clinical adverse reaction |
| Zhang JJ [20] | 2009 | 23 | 22 | | - | - | | - | Lamotrigine | Fluoxetine | | 4W | VAS, clinical adverse reaction |
| Wang YP [21] | 2017 | 90 | 90 | | 64.5±8.3 | 66.3±9.8 | | 109/71 | Serqulin | Vitamin B | | 4W | VAS |

regimens. Network evidence is shown in Fig 3A. The line between the two points represents the evidence of direct comparison between the two drugs. The absence of a line indicates that there is no direct comparison, and the results can be obtained through indirect comparison. The thickness of the line indicates the number of studies using the two drugs in all the included studies. The size of the dot indicates the sample size of the included cases using the drug.

## Inconsistency

After treatment, the four drugs of VAS formed a closed loop, i.e., pregabalin-lamotrigine-carbamazepine-fluoxetine. The overall consistency result showed that P = 0.1782 > 0.05, indicating that there was no total inconsistency. The loop consistency was 0.96, and the lower limit of the 95% CI was 0, indicating that the consistency of each closed loop was good. The inconsistency factor detection and 95% CI between each characteristic cycle and the heterogeneity parameter T2 between cycles are shown in Fig 3Ba. The local consistency results are shown in Table 2. The p value of each comparison was greater than 0.05, indicating good local consistency. After treatment, the three drugs of NRS formed a closed loop, i.e., pregabalin-gabapentin-carbamazepine. The overall consistency result showed that P = 0.3802 > 0.05, indicating that there was no total inconsistency. The loop consistency was 0.79, and the lower limit of the 95% CI was close to 0, indicating that the consistency of each closed loop was good. The inconsistency factor detection and 95% CI between each characteristic cycle and the heterogeneity parameter T2 between cycles are shown in Fig 3Bb. The local consistency results are shown in Table 3. The p value of each comparison was greater than 0.05, indicating good local

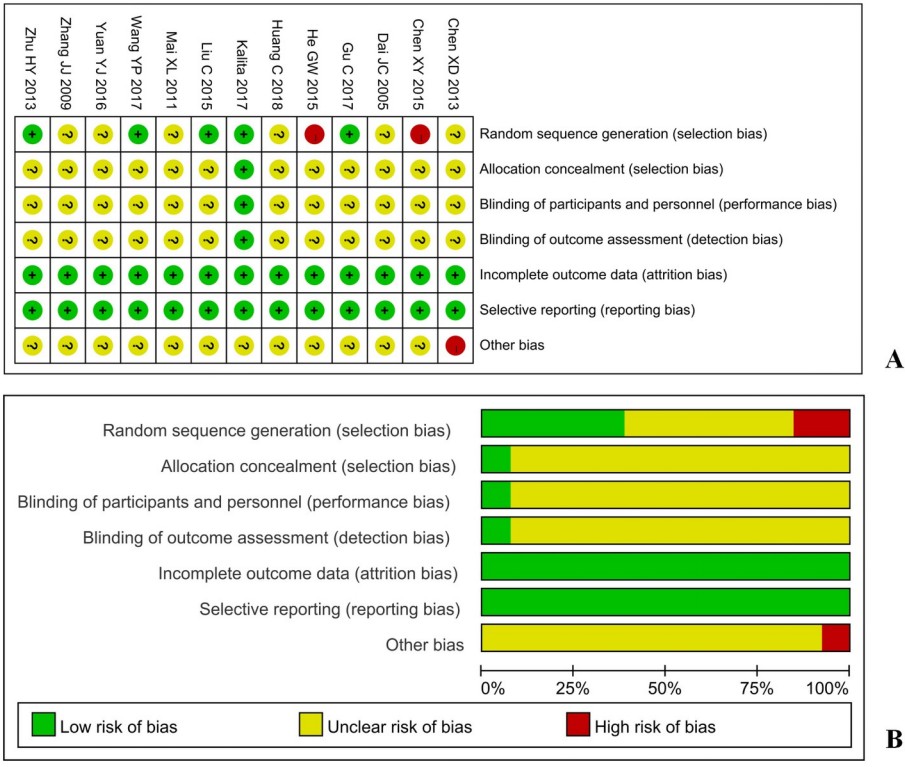

**Fig 2. Risk of bias graph.** A: Risk of bias summary; B: Risk of bias graph.

consistency. HAMD did not form a closed loop, so it was not necessary to conduct the inconsistency test of the closed loop.

## Network meta-analysis

**VAS.** A total of 9 drugs were compared directly and indirectly, and 16 were significantly different. Amitriptyline, vitamin B and carbamazepine reduced VAS scores less than pregabalin. Compared with lamotrigine, vitamin B and carbamazepine decreased VAS scores less. Compared with amitriptyline, carbamazepine, fluoxetine and gabapentin significantly reduced VAS scores, while vitamin B reduced VAS scores less than amitriptyline. Compared with vitamin B, carbamazepine, fluoxetine, gabapentin, duloxetine and serqulin significantly reduced VAS scores. Compared with carbamazepine, fluoxetine and gabapentin significantly reduced VAS scores and had better curative effects. There was no significant difference in other comparisons, as shown in Fig 4.

**NRS.** A total of 3 drugs were compared directly and indirectly, and 2 were significantly different. Compared with pregabalin, gabapentin and carbamazepine decreased NRS significantly less. There was no significant difference in other comparisons, as shown in Fig 5A.

**HAMD.** A total of 4 drugs were compared directly and indirectly, and 2 were significantly different. Compared with pregabalin, gabapentin and amitriptyline decreased HAMD significantly less. There was no significant difference in other comparisons, as shown in Fig 5B.

**SUCRA.** The SUCRA probability ranking results of VAS were as follows: gabapentin (88.3%) > pregabalin (84.5%) > fluoxetine (82.4%) > lamotrigine (57.8%) > duloxetine (51.6%) > serqulin (38.2%) > amitriptyline (25.9%) > carbamazepine (21.2%) > vitamin B (0%). The SUCRA probability ranking results of NRD were as follows: pregabalin (99.5%) >

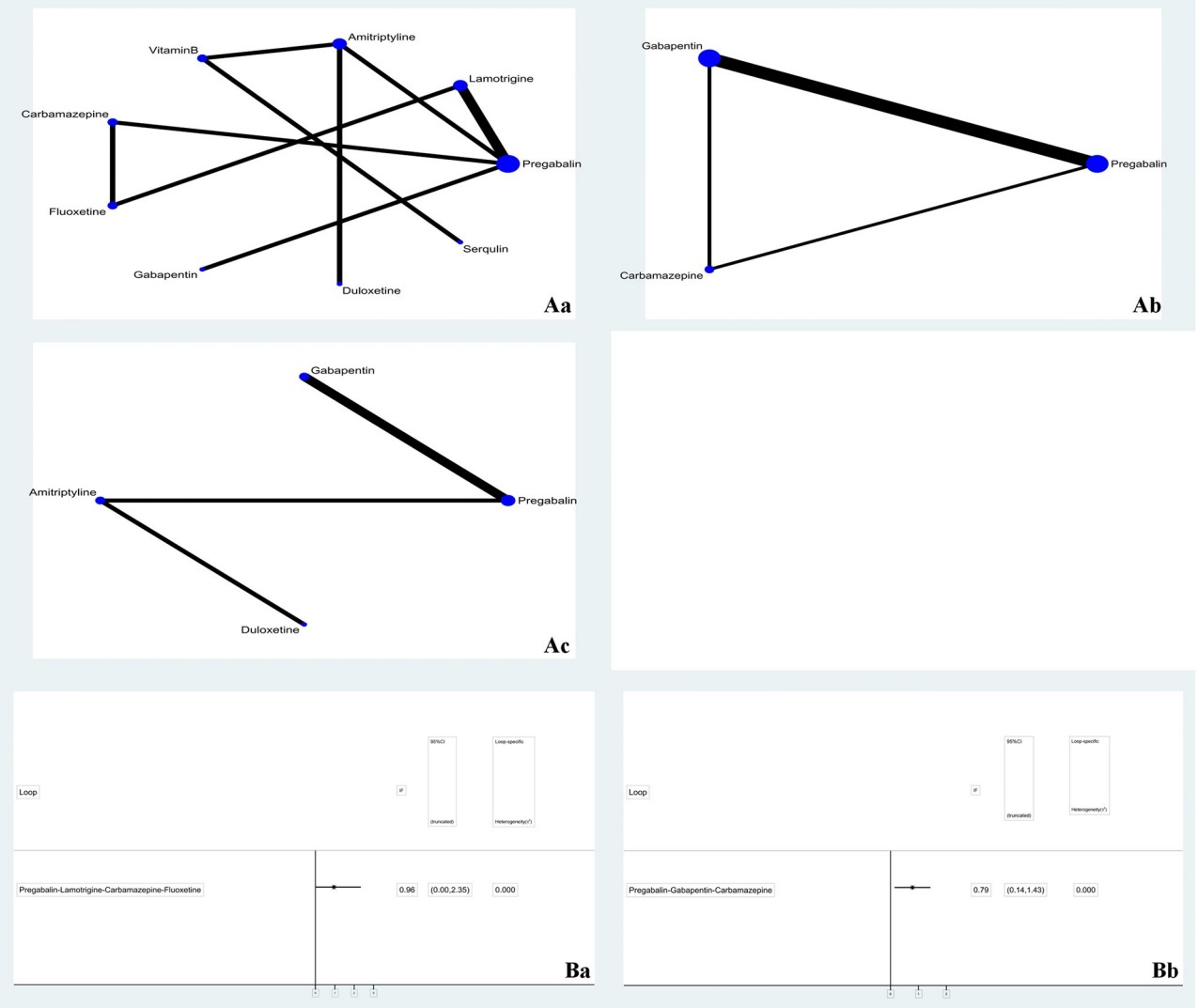

**Fig 3. A.** Network diagram (Aa: VAS; Ab: NRS; Ac: HAMD); **B**. inconsistency (Ba: VAS; Bb: NRS).

gabapentin (49.1%) > carbamazepine (1.3%). The SUCRA probability ranking results of HAMD were as follows: pregabalin (99%) > duloxetine (48.6%) > gabapentin (46.7%) > amitriptyline (5.8%). The cumulative probability ranking diagram is shown in Fig 6A. The larger the area under the curve, the more effective the representation is.

## Publication bias

In this study, the network meta-analysis method was used. The dots of different colors in the funnel chart of VAS after CPSP treatment with 9 types of drugs represent the direct comparison between two pairs of different CPSP treatment, and the number of dots represents the number of studies. Most of the dots in the funnel chart of this study are symmetrically distributed on the vertical line and its two sides. The two sides are basically symmetrical, but there may still be a certain degree of publication bias. There were few direct comparison experiments between the other indicators, so no bias analysis was made. See Fig 6B.

**Table 2. Node splitting result of VAS.**

| Treatment | Direct | Indirect | network | P |
|---|---|---|---|---|
| A VS B | 0.65(0.27,1.02) | -0.31(-1.65,1.03) | 0.53(-0.02,1.07) | 0.178 |
| A VS C* | 1.31(0.45,2.17) | -0.76(-358.79,357.27) | 1.31(0.45,2.17) | 0.991 |
| A VS E | 1.31(0.60,2.02) | 2.27(1.07,3.46) | 1.55(0.86,2.25) | 0.178 |
| A VS G | - | - | - | - |
| B VS F | -0.16(-1.07,0.75) | -1.12(-2.17,-0.06) | -0.52(-1.33,0.28) | 0.187 |
| C VS D* | 2.28(1.77,2.79) | -1.12(-550.47,548.24) | 2.28(1.77,2.79) | 0.990 |
| C VS H* | -0.5(-0.94,-0.06) | -3.12(-1260.73,1254.49) | -0.5(-0.94,-0.06) | 0.997 |
| D VS I* | -2.5(-3.09,-1.91) | -9.68(-1281.71,1262.35) | -2.5(-3.09,-1.91) | 0.991 |
| E VS F | -1.78(-2.46,-1.10) | -0.82(-2.04,0.39) | -1.55(-2.23,-0.87) | 0.178 |

A:Pregabalin; B:Lamotrigine; C: Amitriptyline; D: Vitamin B; E: Carbamazepine; F: Fluoxetine; G: Gabapentin; H: Duloxetine; I: Serqulin. (* all the evidence about these contrasts comes from the trials which directly compare them.)

**Table 3. Node splitting result of VAS.**

| Treatment | Direct | Indirect | network | P |
|---|---|---|---|---|
| A VS B | - | - | - | - |
| A VS C* | 2.84(1.08,4.60) | 1.02(-2.66,4.70) | 2.50(0.99,4.02) | 0.380 |
| B VS C* | 1.16(-0.60,2.92) | 2.98(-0.68,6.65) | 1.50(-0.01,3.02) | 0.380 |

A:Pregabalin; B:Gabapentin; C: Carbamazepine.

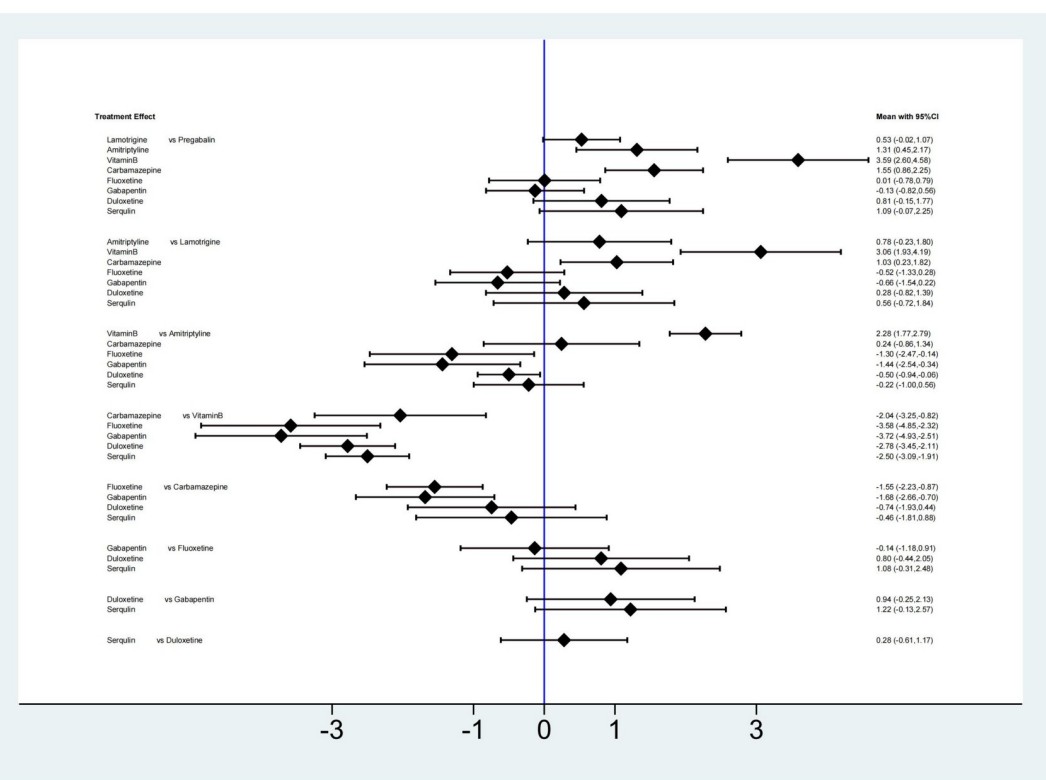

**Fig 4. Network meta-analysis of VAS scores.**

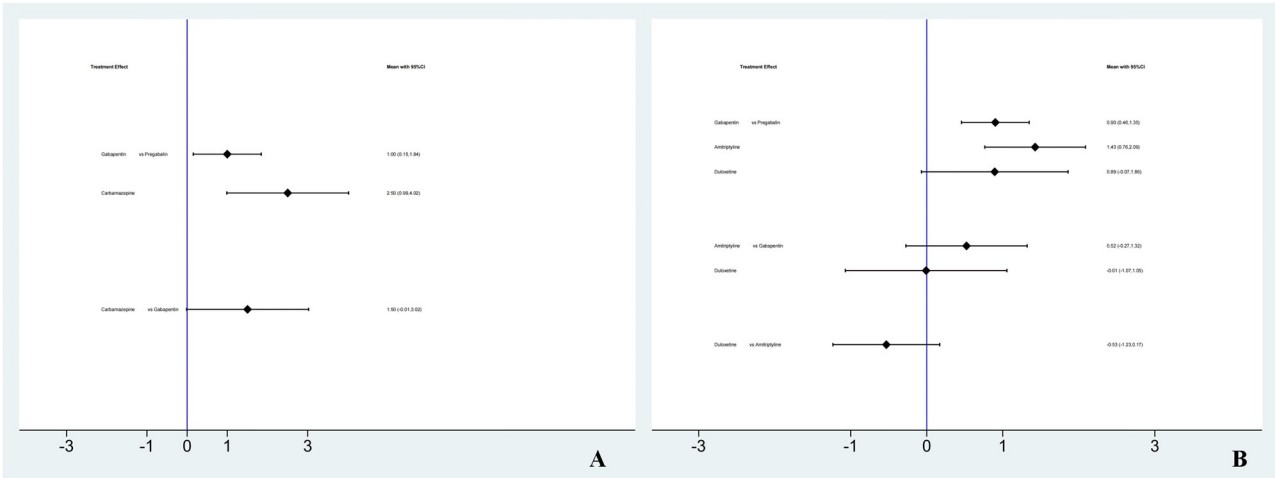

**Fig 5. Network meta-analysis of NRS and HAMD scores.** (A: NRS; B: HAMD).

## Adverse reactions

Of the 13 included studies, 9 mentioned adverse reactions, as shown in Table 1. Nine studies [9, 11, 12, 14–17, 19, 20] reported specific information on adverse reactions. A total of 837 patients with various adverse reactions to agents (pregabalin, gabapentin, lamotrigine, amitriptyline, carbamazepine, fluoxetine, and vitamin B) were included. The adverse reactions were mainly nausea, constipation, dizziness, dry mouth, lethargy, blurred vision, fatigue, peripheral edema, ataxia, leucopenia, and gastrointestinal reactions. See Tables 4 and 5 for specific

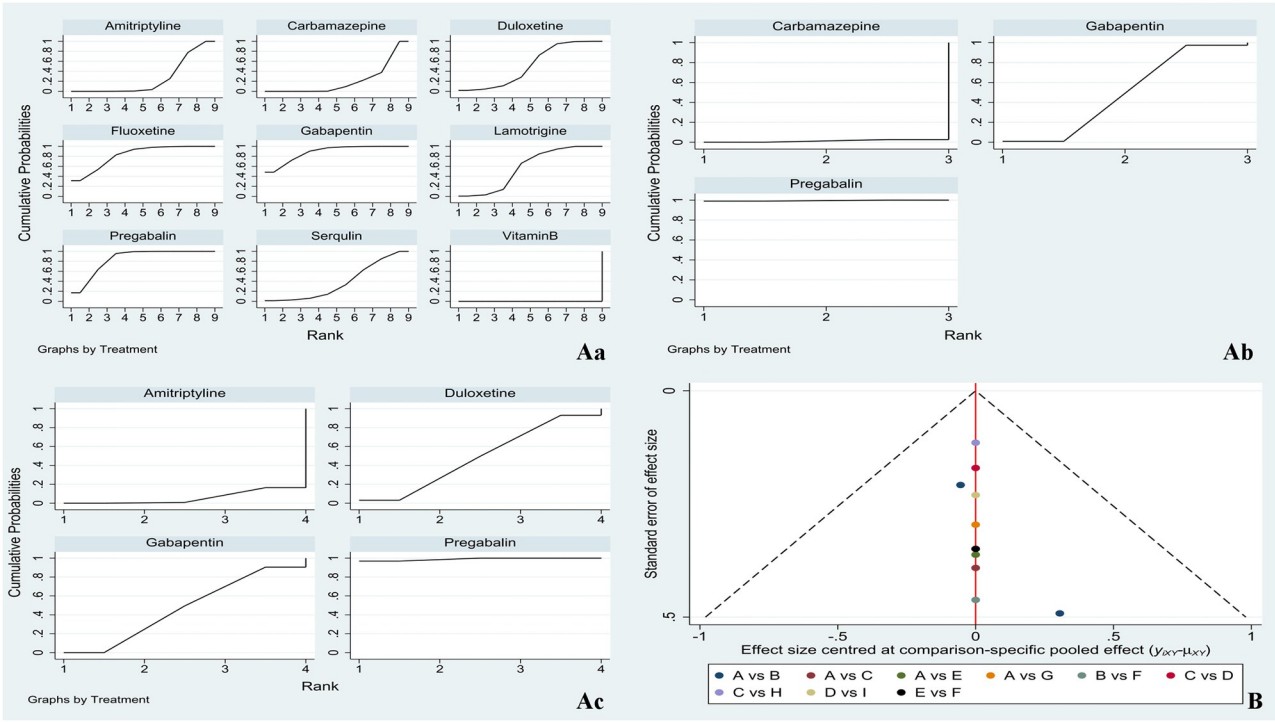

**Fig 6. A**: Cumulative probability ranking plot (Aa: VAS; Ab: NRS; Ac: HAMD); **B**: Funnel diagram of VAS.

**Table 4. Analysis of adverse reactions (number of events).**

| Treatment | Total number | Diarrhea | Weakness | Peripheral edema | Ataxia | Leukocytopenia | Headache | Gastrointestinal reaction | Other |
|---|---|---|---|---|---|---|---|---|---|
| Pregabalin | 305 | - | 4 | 3 | 7 | 3 | - | 2 | 4 |
| Gabapentin | 149 | - | 7 | 5 | 9 | 2 | - | 5 | 5 |
| Lamotrigine | 113 | - | - | 5 | 9 | 9 | 2 | - | 7 |
| Amitriptyline | 99 | 3 | - | - | - | 0 | - | - | 1 |
| Carbamazepine | 84 | - | 8 | 3 | 6 | - | - | 6 | - |
| Fluoxetine | 42 | - | - | - | - | - | 0 | - | - |
| Vitamin B | 45 | 3 | - | - | - | 0 | - | - | 1 |

adverse reactions. The numbers of patients and adverse reactions in the different agent groups were as follows: 1) pregabalin—305 patients had 12 adverse reactions; 2) gabapentin—149 patients had 11 adverse reactions; 3) lamotrigine—113 patients had 10 adverse reactions; 4) amitriptyline—99 patients had 6 adverse reactions; 5) carbamazepine—84 patients had 11 adverse reactions; 6) fluoxetine—42 patients had 2 adverse reactions; 7) vitamin B—45 patients had 5 adverse reactions.

## Discussion

CPSP can affect the quality of life of patients to varying degrees. Although the literature on the prevalence of CPSP is very rich at present, there are relatively few clinical trials of drugs targeting CPSP, and meta-analyses on CPSP are rare [22]. Currently, there are many kinds of drugs used to treat CPSP in the clinic [23]. In routine practice, clinicians have a wide range of treatment options, and they need strong evidence to determine the best treatment scheme for each patient. Therefore, this study combines previous research results with network meta-analysis to compare the efficacy and safety evaluation of different drugs in relationship to CPSP treatment and finally ranks the drugs according to efficacy. From the included literature, 9 drugs were recorded in this study, ad they were subclassified into three categories according to their mechanisms of action: 1) anticonvulsant drugs—pregabalin, lamotrigine, gabapentin, and carbamazepine; 2) antidepressants—amitriptyline, serqulin, fluoxetine, and duloxetine; and 3) nutrition and nerve drugs—vitamin B.

### Clinical efficacy

Currently, VAS and NRS are the most commonly used pain scales in the clinical evaluation of CPSP. The VAS requires patients to mark their pain degree on a Vernier scale. The higher the score, the more obvious the subjective pain of the patient, and the score value can be accurate to the mm. This method was first used in psychological research. Later, for judging the degree of pain [24], the NRS digital grading method expressed different degrees of pain

**Table 5. Analysis of adverse reactions (number of events).**

| Treatment | Total number | Nausea | Constipation | Dizzy | Dry | Urine retention | Pruritus | Drowsiness | Blurred vision |
|---|---|---|---|---|---|---|---|---|---|
| Pregabalin | 305 | 12 | 4 | 38 | 12 | 0 | 0 | 33 | 11 |
| Gabapentin | 149 | 5 | - | 24 | 9 | - | - | 24 | 3 |
| Lamotrigine | 113 | 1 | - | 16 | 6 | - | - | 22 | 16 |
| Amitriptyline | 99 | - | - | 13 | 10 | - | - | 4 | 4 |
| Carbamazepine | 84 | 14 | 14 | 36 | 28 | 4 | 6 | 26 | - |
| Fluoxetine | 42 | 2 | - | 1 | - | - | - | 0 | - |
| Vitamin B | 45 | - | - | 2 | 3 | - | - | - | 1 |

in numbers, and patients were asked to select scores according to their own feelings [25]. The VAS and NRS scores ranged from 0 to 10 most of the time, and the higher the value, the more intense was the patient's pain. The SUCRA ranking results of VAS in this study were as follows: gabapentin > pregabalin > fluoxetine > lamotrigine > duloxetine > serqulin > amitriptyline > carbamazepine > vitamin B. The SUCRA ranking results of NRS in this study were as follows: pregabalin > gabapentin > carbamazepine. The results showed that pregabalin and gabapentin had the best effect in relieving pain in CPSP patients. Gabapentin, as an anticonvulsant, is mainly used to treat partial seizures. Its mechanism of action is still unclear. Some studies have shown that gabapentin may increase GABA levels in different parts of the brain, including the thalamus, and induce glial cells to release GABA to relieve pain. Gabapentin was proven to be an effective and well-tolerated treatment for CPSP patients in a prospective observational study [26]. Pregabalin is a structural analog of the inhibitory neurotransmitter GABA but does not directly bind to GABA receptors. Some studies have shown that the analgesic mechanism of pregabalin may be due to its good fat solubility; it can inhibit a subunit protein of the voltage-dependent calcium ion channels of the central nervous system across the blood-brain barrier and reduce the release of neuro-transmitters and the influx of calcium ions, thereby reducing the release of excitatory neuro-transmitters such as glutamate, norepinephrine and substance P and controlling neuropathic pain [27]. The results of a retrospective analysis in Japan showed that pregabalin can effec-tively relieve pain in patients with CPSP [28]. Gabapentin and pregabalin are also considered first-line drugs for the treatment of central neuropathic pain in the drug treatment guidelines for neuropathic pain developed by the European Union of Neuropathic Associations [29].

HAMD, another secondary indicator in this study, was compiled by Hamilton and is the most widely used scale in the clinical assessment of depression. It has three versions, which include 17 items, 21 items and 24 items. The score is positively correlated with the severity of the patient's condition [30]. HAMD is often used as an auxiliary index in clinical studies to evaluate the impact of pain on the quality of life and mood of CPSP patients. In this study, the SUCRA ranking of HAMD was pregabalin > duloxetine > gabapentin > amitriptyline. The results show that pregabalin is the best intervention measure to improve the HAMD index of CPSP patients. The study shows that patients with chronic pain are more likely to have anxiety and depression [31]. Therefore, the analgesic effect of pregabalin can also help relieve the anxi-ety of CPSP patients.

## Adverse reactions

Nine of the studies included in this study mentioned adverse reactions after the intervention. Considering that individual patients may have several adverse reactions that are not differenti-ated in the basic numbers provided in the literature, it is impossible to calculate the total adverse reaction rate of each drug. At present, it can be determined from the results that the drugs with the most adverse reactions are pregabalin, gabapentin and carbamazepine, and the most common adverse reactions of pregabalin and gabapentin, which showed the best clinical efficacy in this study, are nausea, dizziness, dry mouth and sleepiness. However, due to the small size of the literature sample and differences in treatment course or other factors in the lit-erature, it is difficult to judge the specific causes of adverse reactions and related issues, and further observation is still needed.

## Limitation

To date, this study is the first network meta-analysis on different drug treatments for CPSP. Unlike traditional systematic analysis, network meta-analysis can include direct and indirect

comparisons of different drugs in the study. This study has the following limitations: 1. Most of the included studies were Chinese studies, and the number of studies was small, mainly because there were few English RCTs on drug treatment of CPSP, and most of them were observational studies or retrospective studies without control groups; 2. The overall quality of the literature was poor, and the random method, allocation concealment, blinding method and potential bias were not mentioned in some studies; 3. The baseline included in the study, such as the length of treatment, may increase the possibility of inconsistency and increase the clinical heterogeneity.

## Conclusion

In summary, both anticonvulsants and antidepressants can relieve the pain of CPSP patients to varying degrees. Among them, pregabalin and gabapentin had the most significant effects, while gabapentin and pregabalin had the most adverse effects. However, due to the limitations of this study, the efficacy ranking cannot fully explain the advantages and disadvantages of clinical efficacy and safety and is only for clinical reference. In the future, more multicenter, large-sample, double-blind clinical randomized controlled trials need to be carried out to supplement and demonstrate the results of this study.

## Supporting information

**S1 Checklist. PRISMA NMA checklist of items to include when reporting a systematic review involving a network meta-analysis.**
(DOC)

**S1 Appendix.**
(DOC)

**S1 File.**
(PDF)

**S1 Data.**
(XLSX)

## Author Contributions

**Investigation:** Ruo-Yang Li.

**Writing – original draft:** Ruo-Yang Li.

**Writing – review & editing:** Ke-Yu Chen.

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
