## [Decision Letter · Decision Letter 0]

5 Sep 2022

PONE-D-22-22145Efficacy and safety of different antidepressants and anticonvulsants in central poststroke pain: a network meta-analysis and systematic reviewPLOS ONE

Dear Dr. Li,

Thank you for submitting your manuscript to PLOS ONE. After careful consideration, we feel that it has merit but does not fully meet PLOS ONE’s publication criteria as it currently stands. Therefore, we invite you to submit a revised version of the manuscript that addresses the points raised during the review process.

We look forward to receiving your revised manuscript.

Kind regards,

Walid Kamal Abdelbasset, Ph.D.

Academic Editor

PLOS ONE

Journal Requirements:

Reviewers' comments:

Reviewer's Responses to Questions

**Comments to the Author**

1. Is the manuscript technically sound, and do the data support the conclusions?

Reviewer #1: Yes

Reviewer #2: Partly

2. Has the statistical analysis been performed appropriately and rigorously? 

Reviewer #1: Yes

Reviewer #2: I Don't Know

3. Have the authors made all data underlying the findings in their manuscript fully available?

Reviewer #1: Yes

Reviewer #2: Yes

4. Is the manuscript presented in an intelligible fashion and written in standard English?

Reviewer #1: No

Reviewer #2: Yes

5. Review Comments to the Author

Reviewer #1: The network methodology has been conducted properly. Yet, the language needs to be revised, especially in the introduction, with a more proper word choice.

e.g. in the sentence " Some studies

have shown that CPSP has a certain latency, which can occur at the latest 18 months" can be rephrased in a more scientific language way.

Reviewer #2: This is a valuable research on treatment of Central post stroke pain. The manuscript is presented in an intelligible fashion and written in standard English except for few spilling mistakes in discussion and conclusion sections. In introduction it will be nice to write in short about pathophysiology of central post stroke pain. What about duration of treatment and its relation to efficacy of drugs? Also, what about the site of lesion in stroke and its relation to efficacy of drugs ?

6. PLOS authors have the option to publish the peer review history of their article (what does this mean?). If published, this will include your full peer review and any attached files.

Reviewer #1: **Yes: **Mohamed Mostafa

Reviewer #2: No

---

## [Author Response · Author response to Decision Letter 0]

13 Sep 2022

Response to editor

1.Please ensure that your manuscript meets PLOS ONE's style requirements, including those for file naming. 

Response:The format has been revised according to your requirements.

2.In your Data Availability statement, you have not specified where the minimal data set underlying the results described in your manuscript can be found. PLOS defines a study's minimal data set as the underlying data used to reach the conclusions drawn in the manuscript and any additional data required to replicate the reported study findings in their entirety. All PLOS journals require that the minimal data set be made fully available.

Response: Because this paper is a meta-analysis, all data can be extracted directly from the literature included in this paper. We have uploaded my study’s minimal underlying data set as Supporting Information files.

3.Please review your reference list to ensure that it is complete and correct.

Response:We have reviewed my reference.

Response to reviewers

1. Is the manuscript technically sound, and do the data support the conclusions?

Reviewer #1: Yes

Reviewer #2: Partly

Response: The statistical results in this meta-analysis support this conclusion.

2. Has the statistical analysis been performed appropriately and rigorously?

Reviewer #1: Yes

Reviewer #2: I Don't Know

Response:The statistical methods of this meta-analysis are all operated in strict accordance with the guidelines for meta-analysis

3. Have the authors made all data underlying the findings in their manuscript fully available?

Reviewer #1: Yes

Reviewer #2: Yes

Response: We have uploaded my study’s minimal underlying data set as Supporting Information files.

4. Is the manuscript presented in an intelligible fashion and written in standard English?

Reviewer #1: No

Reviewer #2: Yes

Response: According to your request, I have polished my language to a certain extent.

5. Review Comments to the Author

Reviewer #1: The network methodology has been conducted properly. Yet, the language needs to be revised, especially in the introduction, with a more proper word choice.

e.g. in the sentence " Some studie shave shown that CPSP has a certain latency, which can occur at the latest 18 months" can be rephrased in a more scientific language way.

Response:According to your request, I have polished my language to a certain extent by AJE. We have also revised some words and sentences that are not smooth.

Reviewer #2: This is a valuable research on treatment of Central post stroke pain. The manuscript is presented in an intelligible fashion and written in standard English except for few spilling mistakes in discussion and conclusion sections. In introduction it will be nice to write in short about pathophysiology of central post stroke pain. What about duration of treatment and its relation to efficacy of drugs? Also, what about the site of lesion in stroke and its relation to efficacy of drugs ?

Response: We have revised some grammatical problems in our article by AJE. In the introduction, we added the pathogenesis of CPSP, the common disease site, and the commonly used course of treatment and effect of drugs according to your requirements. However, there are very few relevant studies on the relationship between the lesion site of stroke and the efficacy of drugs, so we did not add this part.

---

## [Decision Letter · Decision Letter 1]

27 Sep 2022

Efficacy and safety of different antidepressants and anticonvulsants in central poststroke pain: A network meta-analysis and systematic review

PONE-D-22-22145R1

Dear Dr. Li,

We’re pleased to inform you that your manuscript has been judged scientifically suitable for publication and will be formally accepted for publication once it meets all outstanding technical requirements.

Kind regards,

Walid Kamal Abdelbasset, Ph.D.

Academic Editor

PLOS ONE

Additional Editor Comments (optional):

Reviewers' comments:

Reviewer's Responses to Questions

**Comments to the Author**

1. If the authors have adequately addressed your comments raised in a previous round of review and you feel that this manuscript is now acceptable for publication, you may indicate that here to bypass the “Comments to the Author” section, enter your conflict of interest statement in the “Confidential to Editor” section, and submit your "Accept" recommendation.

Reviewer #1: All comments have been addressed

Reviewer #2: All comments have been addressed

2. Is the manuscript technically sound, and do the data support the conclusions?

Reviewer #1: Yes

Reviewer #2: Yes

3. Has the statistical analysis been performed appropriately and rigorously? 

Reviewer #1: Yes

Reviewer #2: Yes

4. Have the authors made all data underlying the findings in their manuscript fully available?

Reviewer #1: Yes

Reviewer #2: Yes

5. Is the manuscript presented in an intelligible fashion and written in standard English?

Reviewer #1: Yes

Reviewer #2: Yes

6. Review Comments to the Author

Reviewer #1: I greatly appreciate the work well performed by the authors and the language improvements conducted properly.

Reviewer #2: Thank you for addressing the comments. The manuscript is technically sound with the data support the conclusions . The authors made the data available without restrictions. The manuscript is presented in standard English and in an intelligible fashion.

7. PLOS authors have the option to publish the peer review history of their article (what does this mean?). If published, this will include your full peer review and any attached files.

Reviewer #1: **Yes: **Mohamed Mostafa

Reviewer #2: No

---

## [Editor Report · Acceptance letter]

3 Oct 2022

PONE-D-22-22145R1 

Efficacy and safety of different antidepressants and anticonvulsants in central poststroke pain: A network meta-analysis and systematic review 

Dear Dr. Li:

I'm pleased to inform you that your manuscript has been deemed suitable for publication in PLOS ONE. Congratulations! Your manuscript is now with our production department. 

Kind regards, 

on behalf of

Dr. Walid Kamal Abdelbasset 

Academic Editor

PLOS ONE